# Sentiment Classification using Sentence Embeddings: Exploiting Sentence Transformer Loss Functions
# Journal Submissions

## Abstract

Evaluating customer sentiment plays a critical role in business success. By analyzing customer feedback, companies can swiftly identify expectations, areas for improvement, and pain points related to their products and services. Sentiment analysis, fueled by advances in natural language processing techniques, has become widely accepted for this purpose. In this study, we leverage the popular "Twitter US Airline Sentiment" dataset to develop a sentence transformer architecture based on pre-trained transformer models (MPNet and RoBERTa-Large). We fine-tune the model using appropriate loss functions to generate semantically rich sentence embeddings that are subsequently fed into machine learning algorithms. The resulting hybrid models achieve impressive sentiment prediction performances. Additionally, this study delves into the intricacies of various transformer loss functions that can be applied to fine-tune the sentence transformer model for enhanced sentiment classification performance. Our sentence transformer architecture based on RoBERTa-Large, fine-tuned on CosineSimilarity loss function and combined with XGBoost Classifier, achieved the maximum accuracy of 88.4%, while demonstrating high recall rates even for minority sentiment classes (77.3% for neutral and 83.9% for positive sentiment) without any data augmentation. Furthermore, to evaluate the robustness of our methodology, we also utilized the classic benchmark dataset "IMDB" and achieved an impressive accuracy of 95.9% using a sentence transformer architecture based on RoBERTa-Large, followed by fine-tuning with the CosineSimilarity loss function and combining it with a Support Vector Machine Classifier. We have also done a comparative analysis of our methodology vis-à-vis the advanced Meta-Llama-3-8B large language model in terms of performance, training time and inference time. Our study demonstrates that fine-tuned sentence transformer models can match or even outperform most existing techniques including transformer-based architectures for sentiment classification. They offer the added advantage of reduced computational load and maintain generalizability, even when encountering data that deviates from their fine-tuned training set.

## 1 Introduction

Traditionally, customer feedback surveys have served as a reliable method for companies to gauge customer sentiments. However, these surveys often suffer from low response rates, leading to the voices of majority customers going unheard. Additionally, the time-consuming nature of these surveys impedes swift responses from companies. With the advent of Natural Language Processing (NLP) and Big Data, companies can now gain insights into customer sentiment and identify trends by analyzing social media posts and review websites. In this context, sentiment analysis plays a crucial role. Companies can leverage artificial intelligence to implement real-time analytics of electronic communications from customers, such as comments on social media and online review sites, to understand the sentiments expressed from a consumer perspective. This approach is less intrusive and faster than traditional customer feedback surveys, and it tends to capture unbiased customer opinions more subtly. Consequently, companies can derive quick, actionable insights

and make informed decisions based on the sentiment analysis of customer reviews and feedback. Numerous studies have been conducted to implement sentiment analysis, particularly utilizing transformer models such as BERT (Bidirectional Encoder Representations from Transformers) (Devlin et al., 2019), yielding excellent results. Large language models such as GPT-3 and its successors, have demonstrated superior performance in sentiment classification tasks, especially under fine-tuned settings for domain specific tasks. However, there has been limited focus on leveraging sentence transformer architectures to enhance prediction performance in sentiment analysis. This research aims to investigate whether the combination of sentence transformer models, fine-tuned with suitable transformer loss functions and appropriate machine learning algorithms, can improve the model's prediction performance. This article explores the following research questions:

- How good are sentence embeddings for sentiment classification?

- What are the working theories behind the different transformer loss functions applicable for sentiment classification?

- What is the lift achieved in terms of different performance metrics when using fine-tuned sentence transformer models instead of base sentence transformer models?

This article has been structured as follows: Section 2 depicts related works carried out in this domain; Section3 underlines the data description and the comprehensive design approach followed to build the models; In Section4, the model performance is analyzed; finally Section5 talks about the broader implications and potential future direction of this research . Code is publicly available at https://github.com/***/Sentiment_Classification.

## 2 Understanding the sentiment landscape: SotA Analysis

Numerous research efforts have explored sentiment classification, particularly using the "Twitter US Airline Sentiment" and the "IMDB" datasets. Notable studies include:

- **Capsule network:** In Demotte et al. (2021), authors proposed a novel approach based on capsule network architecture tailored for social media content analysis. Remarkably, the proposed approach achieved reasonably good performance even without relying on any linguistic resources, demonstrating their effectiveness in sentiment analysis. The proposed architectures achieved an accuracy of 82.04% on the US Airline dataset. This result highlighted significant accuracy enhancements in text processing for social media content analysis and at the same time offered a fresh perspective for sentiment analysis in the dynamic landscape of social media.

- **Affection Driven Neural Networks:** In Xiang et al. (2020), the authors focused on one of the key challenges being faced by the deep neural networks for sentiment analysis which is effectively incorporating external sentiment knowledge. In this work, the authors proposed an innovative approach called "affection-driven neural networks" that leverages affective knowledge. Affective knowledge is obtained in the form of a lexicon based on the Affect Control Theory. This lexicon represents affective attributes using three-dimensional vectors: Evaluation, Potency, and Activity (EPA). The EPA vectors are mapped to an affective influence value and integrated into LSTM (Long Short-term Memory) models. This integration allows the neural network to highlight affective terms during sentiment analysis. The proposed approach consistently outperformed conventional LSTM models by 1.0% to 1.5% in terms of accuracy across three large benchmark datasets. For the US airline dataset, the proposed neural network architecture comprising of LSTM-AT (LSTM with attention mechanism to re-weight important words before the fully connected layer) with Evaluation achieved the highest accuracy of 82.8% whereas for IMDB, BiLSTM-AT i.e. BiLSTM with attention mechanism and Potency achieved the highest accuracy of 82.6%.

- **RoBERTa-LSTM:** In Tan et al. (2022)the authors combined the RoBERTa model with Long Short-Term Memory (LSTM) networks. RoBERTa (Robustly Optimized BERT Approach) (Liu et al., 2019) is a variant of BERT optimized for sequence-to-sequence modeling. Like BERT, RoBERTa is

a transformer-based language model that uses self-attention to process input sequences and generate contextualized representations of words in a sentence. It generates word embeddings effectively. One crucial difference between RoBERTa and BERT is that RoBERTa was trained on a much larger dataset. Moreover, RoBERTa uses a dynamic masking technique during training that helps the model learn more robust and generalizable representations of words. RoBERTa is utilized to convert words into a compact and meaningful word embedding space, while LSTM effectively captures long-distance contextual semantics. This hybrid model addresses challenges such as lexical diversity, imbalanced datasets, and long-distance dependencies in text. The experimental results showed an accuracy of 85.9% (without data augmentation) for the Twitter US Airline Sentiment dataset and 93.0% for the IMDB dataset.

- **Hybrid BERT Models:** In Talaat (2023), the author proposed a hybrid approach that combines BERT with Bidirectional Long Short-Term Memory (BiLSTM) and Bidirectional Gated Recurrent Unit (BiGRU) algorithms. The author created hybrid deep learning models by stacking two versions of BERT (RoBERTa & DistilBERT) with BiLSTM and BiGRU layers. The proposed architecture of RoBERTa with BiGRU layers yielded the highest accuracy of 86% for the US airlines dataset.

- **Flan 137B:** In Wei et al. (2022), the authors of this paper introduced FLAN, a 137B parameter language model fine-tuned on over 60 NLP datasets using natural language instructions, which significantly enhances zero-shot performance on unseen tasks. FLAN outperformed the 175B GPT-3 in zero-shot settings on majority of the evaluated datasets and even exceeded GPT-3's few-shot performance on several tasks such as ANLI (Adversarial Natural Language) and RTE (Recognizing Textual Entailment). This model achieved a zero-shot accuracy of 94.3% and few-shot accuracy of 95.0% on the IMDB dataset.

- **XLNet:** In Yang et al. (2019) , the authors proposed XLNet model which addresses the pretrain-finetune discrepancy and dependency issues in BERT by using an autoregressive approach. This model achieved the state-of-the-art performance with an error rate of 3.2% on the IMDB dataset.

Table 1 displays performances of the existing techniques vis-à-vis our technique on the Twitter US Airline Sentiment dataset (TAS) and IMDB dataset.

Table 1: Comparison of Accuracies of existing SotA techniques with our technique

| Model used | TAS | IMDB |
|---|---|---|
| RoBERTa (Kumawat et al., 2021) | 80.8% | |
| BERT (Kumawat et al., 2021) | 81.2% | |
| Capsule Network (Demotte et al., 2021) | 82.0% | |
| LSTM-AT / BiLSTM-AT (Xiang et al., 2020) | 82.3% | 82.6% |
| RoBERTa-LSTM (Tan et al., 2022) | 85.9% | 93.0% |
| RoBERTa with BiGRU layers (Talaat, 2023) | 86.0% | |
| FLAN-zeroshot (Wei et al., 2022) | | 94.3% |
| FLAN-fewshot (Wei et al., 2022) | | 95.0% |
| Our Approach - Fine-tuned Sentence Transformer (MPNet) + LGBMC | 86.5% | |
| Our Approach - Fine-tuned Sentence Transformer (RoBERTa-Large) + XGBoost | **88.4**% | |
| Our Approach - Fine-tuned Sentence Transformer (RoBERTa-Large) + SVM | | **95.9**% |
| XLNet (Yang et al., 2019) | | _**3.2**_% (Error rate) |

[a] Note 1: Bold figures indicate performance of our best performing model.
[b] Note 2: Bold and underlined figures indicate performance by the best SotA model.

Table 2: Data distribution

| #Records | #Positive Tweets | #Neutral Tweets | #Negative Tweets |
|---|---|---|---|
| 14,640 (TAS) | 2,363 (16%) | 3,099 (21%) | 9,178 (63%) |
| 1630 (Apple) | 143 (9%) | 686 (42%) | 801 (49%) |
| 50,000 (IMDB) | 25,000 (50%) | | 25,000 (50%) |
| 400,000 (Yelp - test dataset) | 200,000 (50%) | | 200,000 (50%) |

## 3 Methodology

### 3.1 Data

In this study, the first dataset used is a very popular one, taken from Kaggle's Twitter US Airline Sentiment (CrowdFlower, 2015) which consists of scraped tweets from 2015 Twitter for 6 major airlines operating in US with 'positive', 'negative' and 'neutral' labels. This dataset was originally released by CrowdFlower in 2015 and comprises of 14,640 tweets spread across three sentiment classes, positive, negative, and neutral and was labelled manually. However, this dataset is imbalanced as evident from table 2 with maximum records having negative sentiment. The tweets in the dataset belong to six American airlines which are American, Delta, Southwest, United, US Airways, and Virgin America. There are 15 columns in the dataset, however, for this study, only the text and airline sentiment columns have been used because the research focuses solely on textual data and its corresponding sentiment label. This is to ensure that the model ingests only the relevant data, avoiding the inclusion of any unrelated information. The second dataset used is the IMDb dataset (Maas et al., 2011) which is widely recognized as a benchmark for sentiment analysis. It consists of 50,000 movie reviews taken from IMDB, with an equal split of positive and negative reviews. To further evaluate the generalization ability of our transformer models fine-tuned on the aforementioned datasets, we have also included two additional independent datasets: Twitter Apple Sentiment (Apple) (CrowdFlower) and Yelp-2 (only test set) (Zhang & Acharki, 2022).

### 3.2 Design Approach

- Transformer Architecture: A sentence transformer was created from a pretrained transformer model (MPNet and RoBERTa-Large) by performing pooling on the token embeddings. This novel concept of generating sentence embeddings using siamese and triplet network structures was introduced in Reimers & Gurevych (2019).

- Fine Tuning: The Sentence Transformer was further finetuned on the training dataset using applicable loss functions.

- Text Vectorization: Converted the field 'text' of the training data to numerical representation using sentence transformer embeddings.

- Model Building and Prediction: Embeddings were used as attributes to create a hybrid machine learning model based on machine learning algorithms. The hybrid model predicted the sentiment on unseen text.

Figure 1 describes the entire process flow briefly.

### 3.3 Proposed Model Architecture and Methodology

#### 3.3.1 Underlying Sentence Transformer Model

To begin with, the first underlying base transformer model that has been used is MPNet as it combines the strength of masked language modeling adopted in BERT and permuted language modeling adopted in XLNet (Yang et al., 2019). This MPNet model was proposed in Song et al. (2020) where the authors

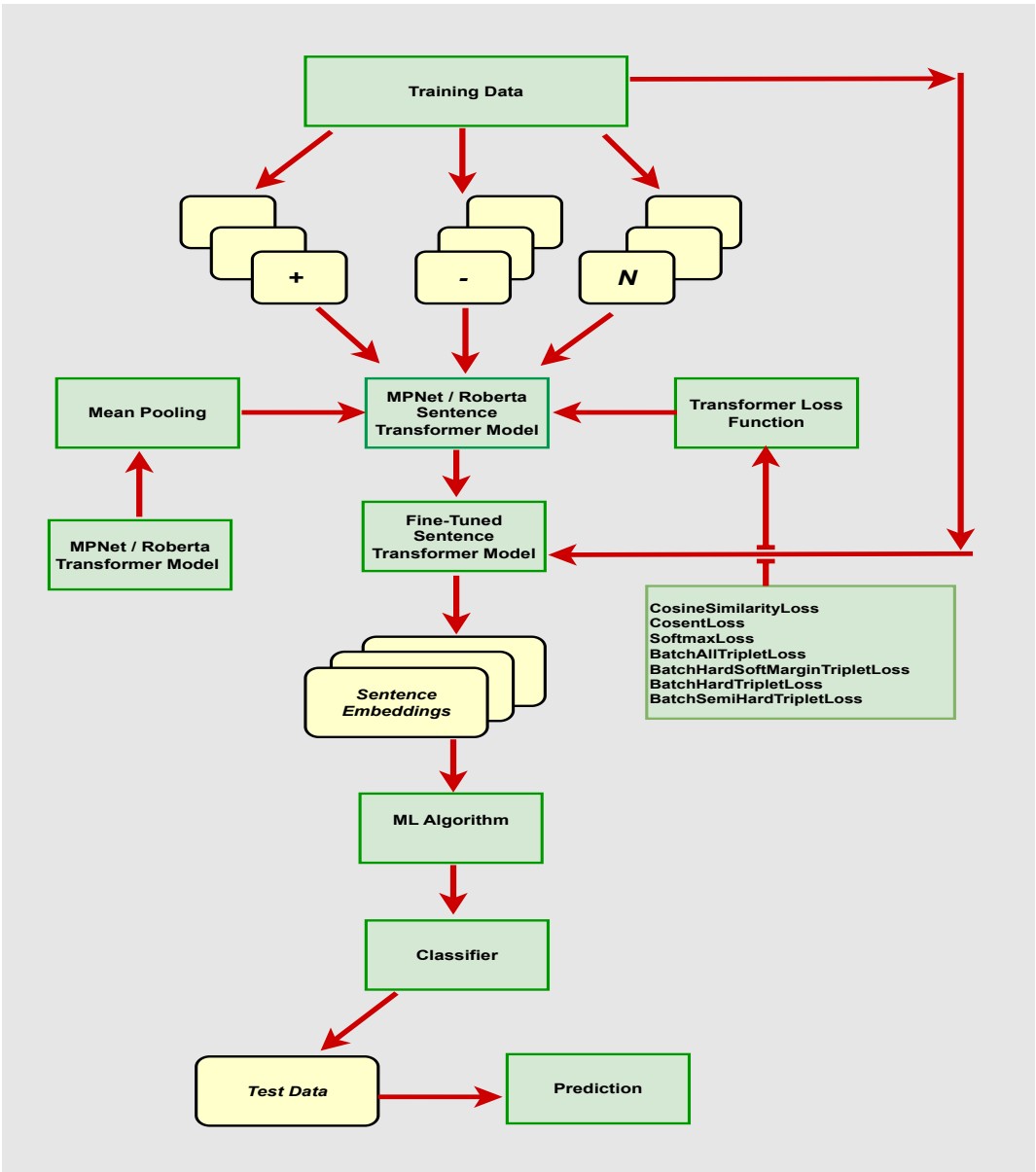

Figure 1: Process flow diagram

identified drawbacks for both BERT and XLNet. Though BERT has been enormously successful for the most common natural language processing tasks like sentiment classification and named entity recognition, yet it overlooks the dependency among predicted tokens during pre-training. XLNet introduced permuted language modeling to address this issue but suffers from position discrepancy between pre-training and fine-tuning. MPNet not only leverages the dependency among predicted tokens like XLNet but it also takes auxiliary position information as input, allowing the model to see the full sentence and reducing position discrepancy. By combining these features, MPNet delivers superior performance when compared to its peers like BERT.

The second underlying base transformer model that has been used is RoBERTa-Large which is a transformer-based language model developed by Meta AI. It is an enhanced version of BERT, trained on a larger dataset

with larger batch sizes and longer sequences compared to BERT, thus enhancing its abilities to learn more from the data.

As a sentence moves through a transformer model, it creates word or token embeddings. These embeddings are averaged using a mean pooling layer to form a fixed-length sentence embedding. This approach captures the sentence's overall meaning with less detail, resulting in a sentence transformer. Thus, sentence transformers reduce computational load by working at the sentence level while achieving similar results (with fine-tuning) to models like BERT or RoBERTa

Figure 2 depicts the creation of sentence transformer architecture briefly.

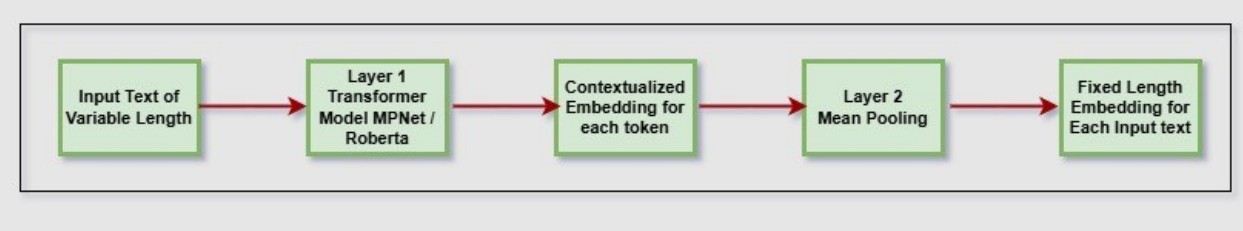

Figure 2: Sentence transformer creation process

### 3.3.2 Transformer Loss Functions

Now the above sentence transformer model will be fine-tuned on the training dataset with the help of suitable loss functions for an enhanced performance on the target task. Selection of appropriate loss function is critical towards model's performance. Unfortunately, there is no fixed recipe to determine the most fitting loss function. However, it largely depends on the structure of the data and the target task. Since the dataset used here is labelled, the loss functions relevant for supervised learning are discussed here. Details of all the loss functions can be found in Reimers & Gurevych. Principally, the objective of the loss function is to optimize such that texts with same labels are as close as possible in the vector space whereas the texts with different labels lie farthest from each other. The loss functions that have been used in the architecture are as follows:

1. **CosineSimilarityLoss**
2. **Cosine Sentence Loss (CosentLoss)**
3. **SoftmaxLoss**
4. **BatchAllTripletLoss**
5. **BatchHardSoftMarginTripletLoss**
6. **BatchHardTripletLoss**
7. **BatchSemiHardTripletLoss**

**CosineSimilarityLoss**: In order to finetune a sentence transformer model on CosineSimilarityLoss, the model needs to be trained that a pair of texts from the training data has a predefined degree of cosine similarity. Therefore, each training example should consist of a pair of texts from the data along with a label indicating their similarity score that allows the model to understand how similar the two texts are. So, the next step involves converting the dataset into a format that can be ingested by the sentence transformer model. The model cannot accept raw text. Hence each example must be converted to a sentence_transformers.InputExample class and then to a torch.utils.data.DataLoader class to train the model.

Below is a toy example for demonstration purpose showing how to fine tune the model on cosine similarity loss.

```
train_examples = [
    InputExample(texts=[ postext1, postext2], input_label=1),
    InputExample(texts=[ postext1, negtext1], input_label =0),
    InputExample(texts=[ postext1, neutext1], input_label =0.5),
]
```

Here the model is being taught that the similarity score between the texts in the first pair is 1 since they belong to the same class of positive sentiments. Similarity score between the texts in the second pair is set to 0 so that the model understands that a positive sentiment text has no similarity with a negative sentiment text. Finally, the third pair has the input label set to 0.5 to inform the model that a positive sentiment text is somehow related to a neutral sentiment text but not closely enough. Then the model encodes each pair of texts, using the underlying sentence transformer model into fixed length embeddings, say $vect_1$ and $vect_2$. Then it computes the cosine similarity between the two, say $sim(vect_1, vect_2)$. Subsequently, it uses mean squared error (MSE) as the loss function to compare $sim(vect_1, vect_2)$ with the input label and while minimizing MSE, it aims to minimize the L2 norm of the error given as:

$$Loss = \|input\_label - sim(vect_1, vect_2)\|_2 \tag{1}$$

Here input label is the target similarity score in the range [0,1] for each pair of sentences. So similar pair of sentences belonging to the same class should be set for a similarity score closer to 1 whereas dissimilar pair of sentences belonging to different classes should be set for a lower similarity score. One interesting aspect here is although the cosine similarity ranges between [-1,1], the label or the target similarity score has been deliberately set in the range [0,1] since for every normalized vector (p), there exists exactly one normalized vector (q) such that the cosine similarity between (p) and (q) is -1. However, there are infinite vectors for which the cosine similarity between (p) and each of those vectors is 0. Consequently, using a loss function that only becomes 0 when the cosine similarity between two vectors is -1 is undesirable.

**Cosine Sentence Loss (CoSENT)**: CoSENT loss function, introduced in Jianlin (2022) expects each of the input examples consists of a pair of texts and a target label, representing the expected pairwise similarity score between the texts in the pair. The generic loss function for CoSENT loss is defined as

$$Loss = log(1 + \sum_{sim(i,j)>sim(k,l)} e^{\lambda(s(k,l)-s(i,j))}) \tag{2}$$

where (i,j) and (k,l) are any random input pairs of texts from the training data set such that pairwise cosine similarity between(i,j) , say s(i,j) is always greater than the pairwise cosine similarity between(k,l), say s(k,l). The summation is over all possible pairs of input pairs in the batch that match this condition and hence this loss function is also applicable for multi-class classification problems as long as there is an ordinal relation among the different classes. As evident, in order to minimize the loss, the expression $\sum_{sim(i,j)>sim(k,l)} e^{\lambda(s(k,l)-s(i,j))}$ needs to be minimized which can be achieved only by pushing s(i,j) up and pushing s(k,l) down. Here $\lambda$ is a hyperparameter for scaling the pairwise cosine similarity scores.

Below is a toy example for demonstration purpose showing how to fine tune the model on cosent loss.

```
train_examples = [
    InputExample(texts=[ postext1, postext2], input_label=1),
    InputExample(texts=[ postext1, negtext1], input_label =-1),
    InputExample(texts=[ postext1, neutext1], input_label =0),
]
```

Here the model is being taught that the expected pairwise similarity score between the texts in the first pair is 1 since they belong to the same class of positive sentiments. Expected pairwise similarity score between the texts in the second pair is set to -1 so that the model understands that a positive sentiment text has opposite meaning with respect to a negative sentiment text. Finally, the third pair has the input label set to 0 to inform the model that a positive sentiment text is not related to a neutral sentiment text and their embeddings are orthogonal to each other. Unlike CosineSimilarityLoss, here the target similarity score is in the range [-1,1] for each pair of sentences since pushing the similarity score towards 1 for the pair with

similar texts (positive and positive) and towards -1 for the pair with dissimilar texts (positive and negative) will ensure minimum loss.

**SoftmaxLoss**:This loss was introduced by the authors in Reimers & Gurevych (2019). For each pair of texts under consideration, it concatenates the corresponding sentence embeddings, say $vect_a$ and $vect_b$ with the element-wise difference $|vect_a - vect_b|$ and multiplies it with the trainable weight $W_t \in R^{3n*k}$. Then it adds a softmax classifier to it. So, the final output is given by:

$$Output = softmax(W_t(vect_a, vect_b, |vect_a - vect_b|)) \tag{3}$$

Here n denotes the dimension of the sentence embeddings, k denotes the number of labels or classes and the loss optimized is the cross-entropy loss.

**Triplet Loss**: This concept was introduced in Schroff et al. (2015). Here triplet refers to three entities – one anchor text, one positive text belonging to the same class of anchor text and one negative text belonging to a different class than that of the anchor text. Let us say the corresponding sentence embeddings are a, p and n respectively. The objective of the loss function is to push down the distance between a and p, say d(a,p) towards 0 and simultaneously push up the distance between a and n, say d(a,n) such that $d(a, n) > d(a, p) + \alpha$ where $\alpha$ is a hyperparameter. Hence the loss function can be defined as:

$$Loss = max(d(a, p) - d(a, n) + \alpha, 0) \tag{4}$$

As evident, $d(a, p) - d(a, n) + \alpha$ must always be negative to ensure we get 0 loss. The goal of the triplet loss is to make sure that:

- Two texts from the same class have their embeddings close together in the embedding space

- Two texts from different class have their embeddings far apart by some margin.

There are three types of triplets as follows:

- **Easy Triplets**: triplets with 0 loss, because $d(a, p) + \alpha < d(a, n)$

- **Hard Triplets**: triplets where the negative entity is closer to the anchor than the positive entity, i.e. $d(a, n) < d(a, p)$

- **Semi Hard Triplets**: triplets where the negative entity is not closer to the anchor than the positive entity, but have positive loss, i.e. $d(a, p) < d(a, n) < d(a, p) + \alpha$

$\alpha$ is more commonly referred as the margin. The distance function used here is the Euclidean distance.

**BatchAllTripletLoss** tries to minimize the loss for all valid triplets from the training data.

**BatchHardTripletLoss** uses only the hardest positive and negative samples, rather than all possible, valid triplets from the training data. So, for each anchor, it gets the hardest positive and hardest negative to form a triplet.

**BatchHardSoftMarginTripletLoss** also uses only the hardest positive and negative samples from the training data. However, it does not require a margin.

**BatchSemiHardTripletLoss** uses only semi-hard triplets out of all valid triplets from the training data.

Figure 3 graphically demonstrates the differences among hard, semi-hard and easy triplets along the embedding space.

A triplet (a, p, n) is valid if a, p, n are distinct and label[a] = label[p] and label[a] $\neq$ label[n]. Here labels are represented as integers, where the same label corresponds to sentences from the same class. Additionally, the training dataset should include a minimum of two examples per label class.

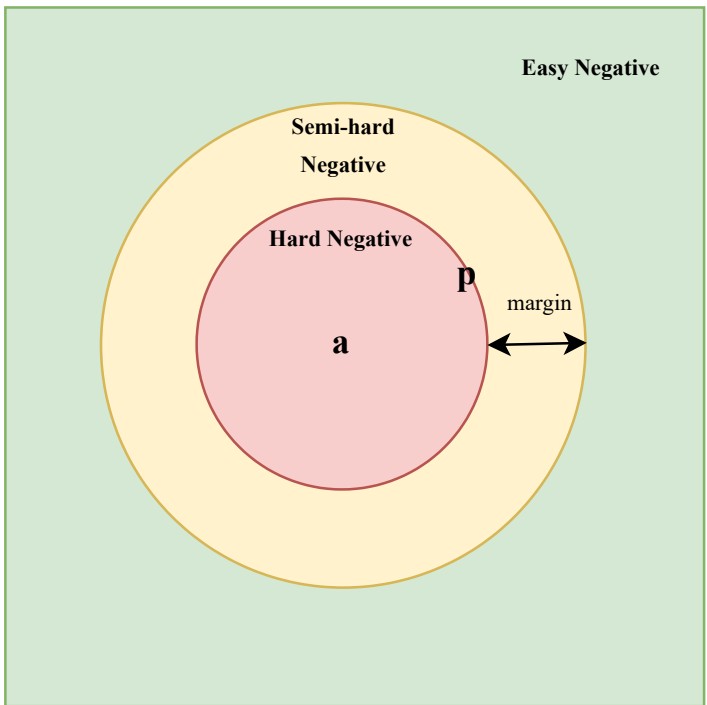

Figure 3: Triplets along embedding space.

### 3.3.3 ML Algorithms

For design experiment modeling, the fine-tuned models were used in combination with following popular ML algorithms for classification:

- **LightGBM (LGBM):** Authors in Ke et al. (2017) efficiently addresses the challenges posed by high feature dimension and large data size in Gradient Boosting Decision Tree by proposing LightGBM which can significantly improve efficiency and scalability. LightGBM stands for Light Gradient-Boosting Machine which is a fast, distributed, gradient boosting framework based on decision tree algorithm and is used for classification and other machine learning problems. It leverages gradient boosting to construct a strong learner by sequentially adding weak learners in a gradient descent manner. Unlike other boosting algorithms, LightGBM splits the tree leaf wise with the best fit i.e. it chooses the leaf that yields the largest decrease in loss, resulting in efficient tree construction. It also uses histogram-based decision tree learning algorithm which buckets continuous attribute values into discrete bins resulting in faster training and lower memory usage. Hence the word "Light" comes into play.

  LightGBM utilizes two novel techniques:

  **Gradient-based One-Side Sampling (GOSS):** This technique enables LightGBM to put emphasis on data instances with larger gradients, which are more significant for computing information gain, leading to accurate gain estimation with reduced data size.

**Exclusive Feature Bundling (EFB):** EFB bundles mutually exclusive features to reduce the number of features without significantly affecting the accuracy of split point determination, thus enhancing efficiency.

- **XGBoost:** XGBoost which stands for eXtreme Gradient Boosting, is a powerful machine learning tool known for its efficiency, speed, and accuracy. This technique was introduced in Chen & Guestrin (2016). Like LGBM, XGBoost is based on based on decision tree algorithms and leverages gradient boosting to construct a strong learner by sequentially adding weak learners in a gradient descent manner. The paper proposes a novel sparsity-aware algorithm for sparse data. It also introduces a weighted quantile sketch for approximate tree learning. Unlike LGBM, XGBoost builds trees level-wise (depth-wise), expanding the tree layer by layer, adding new models to correct errors made by previous ones, which leads to higher memory usage compared to LGBM.

- **Support Vector Machine (SVM) Classifier:** The SVM classifier is a popular machine learning algorithm for classification tasks, renowned for its capability to identify the optimal decision boundary between classes by maximizing the margin separating them. We have leveraged this powerful yet simple algorithm to achieve impressive accuracy on the IMDB test dataset.

## 4 Results

### 4.1 Model Evaluation Parameters

The training data for the Twitter US Airline Sentiment dataset comprises of 80% of the original data and the model predicts on the test data comprising of the remaining 20%. IMDB dataset is equally split into training and test segments.

The model fine-tuned and trained on the Twitter US Airline Sentiment training dataset is also evaluated on the entire Twitter Apple Sentiment dataset to assess its generalization ability. Similarly, the model fine-tuned and trained on the IMDB training dataset is tested on the Yelp test dataset.

Performance evaluation parameters used in this study are Precision, Recall, F1 and Accuracy. Precision, Recall and F1 have been calculated for each class. Hence True Positive, True Negative, False Positive and False Negative also need to be understood with context to each individual class. For example, when calculating the metrics for positive sentiment class, any review having label other than positive sentiment will be considered as a negative instance. Similarly, when calculating the metrics for negative sentiment class, any review having label other than negative sentiment will be considered as a negative instance and so on.

True Positive(TP): The number of instances where the classification system correctly predicts a positive sentiment as a positive one or a negative sentiment as a negative one or a neutral sentiment as a neutral one.

True Negative(TN): The number of instances correctly predicted as negative instances.

False Positive(FP): The number of instances incorrectly predicted as positive instances.

False Negative(FN): The number of instances incorrectly predicted as negative instances.

Precision : Precision is defined as the ratio of correctly classified positive instances to the total number of samples predicted as positive.

$$\text{Precision} = \frac{TP}{TP + FP} \tag{5}$$

Recall: Recall, also known as sensitivity measures how many of the actual positive instances were correctly predicted. This metric is particularly important in determining how well the model is predicting the minority class in an imbalanced dataset

$$\text{Recall} = \frac{TP}{TP + FN} \tag{6}$$

F1 score: F1 score is the harmonic mean of Recall and Precision, combining both metrics into a single value. It is the most used metric after accuracy. It balances precision and recall. F1 score manages the trade-off

between recall and precision.

$$F1 = \frac{2 * (Precision * Recall)}{Precision + Recall} \tag{7}$$

Accuracy: It is the ratio of correct classifications to total predictions given by the model. Accuracy is a good metric to use for sentiment classification when the sentiment classes are balanced.

$$Accuracy = \frac{CorrectPredictions}{TotalPredictions} \tag{8}$$

Table 3 provides a detailed breakdown of the results.

Table 3: Results summary

| Model | Accuracy | Negative | | | Neutral | | | Positive | | |
|---|---|---|---|---|---|---|---|---|---|---|
| | | Precision | Recall | F1 | Precision | Recall | F1 | Precision | Recall | F1 |
| **Twitter US Airline Sentiment** | | | | | | | | | | |
| Baseline MPNet ST + LGBM | 82.9% | 86.7% | 93.0% | 89.8% | 68.3% | 61.9% | 65.0% | 84.4% | 70.8% | 77.0% |
| Baseline MPNet ST + XGB | 83.5% | 86.8% | 94.1% | 90.3% | 70.7% | 61.6% | 65.9% | 84.2% | 71.0% | 77.1% |
| MPNet ST + CosineSimilarityLoss + LGBM | 86.5% | 92.0% | 91.5% | 91.7% | 72.0% | 74.4% | 73.2% | 84.9% | 82.9% | 83.9% |
| MPNet ST + CosineSimilarityLoss + XGB | 86.2% | 91.4% | 91.8% | 91.6% | 72.0% | 72.1% | 72.0% | 84.5% | 83.1% | 83.8% |
| Baseline RoBERTa-Large ST + LGBM | 83.9% | 87.2% | 94.1% | 90.5% | 71.5% | 62.4% | 66.7% | 83.8% | 72.1% | 77.5% |
| Baseline RoBERTa-Large ST + XGB | 84.2% | 87.6% | 94.0% | 90.7% | 71.8% | 62.4% | 66.8% | 84.5% | 74.8% | 79.4% |
| RoBERTa-Large ST + CosineSimilarityLoss + LGBM | 88.0% | 93.3% | 92.9% | 93.1% | 75.4% | 76.9% | 76.1% | 84.7% | 83.9% | 84.3% |
| **RoBERTa-Large ST + CosineSimilarityLoss + XGB**[1] | **88.4%** | 93.2% | 93.3% | 93.3% | 76.5% | 77.3% | 76.9% | 85.4% | 83.9% | 84.7% |
| **Twitter Apple Sentiment** | | | | | | | | | | |
| RoBERTa-Large ST + CosineSimilarityLoss + XGB[2] | 88.5% | 91.2% | 90.2% | 90.7% | 87.6% | 90.6% | 89.1% | 79.5% | 67.8% | 73.2% |
| **IMDB** | | | | | | | | | | |
| Baseline RoBERTa-Large ST + LGBM | 90.2% | 90.4% | 90.0% | 90.2% | | | | 90.1% | 90.4% | 90.2% |
| Baseline RoBERTa-Large ST + XGB | 90.9% | 91.1% | 90.7% | 90.9% | | | | 90.8% | 91.2% | 91.0% |
| Baseline RoBERTa-Large ST + SVM | 90.5% | 91.1% | 89.9% | 90.5% | | | | 90.0% | 91.2% | 90.6% |
| RoBERTa-Large ST + CosineSimilarityLoss + LGBM | 95.6% | 94.7% | 96.6% | 95.6% | | | | 96.6% | 94.6% | 95.6% |
| RoBERTa-large ST + CosineSimilarityLoss + XGB | 95.5% | 94.9% | 96.3% | 95.6% | | | | 96.2% | 94.8% | 95.5% |
| **RoBERTa-Large ST + CosineSimilarityLoss + SVM**[3] | **95.9%** | 96.4% | 95.4% | 95.9% | | | | 95.4% | 96.4% | 95.9% |
| **Yelp** | | | | | | | | | | |
| RoBERTa-Large ST + CosineSimilarityLoss + SVM[4] | 94.8% | 96.6% | 92.8% | 94.7% | | | | 93.0% | 96.8% | 94.9% |

As evident from Table 3, for the Twitter US Airline Sentiment, among the baseline sentence transformer models without any fine-tuning, RoBERTa sentence transformer model in conjunction with XGBoost achieves

---

[1]our best performing fine-tuned model of Twitter US Airline Sentiment data

[2]our best performing fine-tuned model of Twitter US Airline Sentiment data tested on Twitter Apple data

[3]our fine-tuned model of IMDB data

[4]our fine-tuned model of IMDB data tested on Yelp data

the highest overall accuracy of 84.2% and high precision, recall and F1 scores for the negative sentiment class. This indicates that the sentence embeddings generated by the baseline sentence transformer model without any fine-tuning are semantically rich. However, for the minority classes of positive and neutral sentiments, the performance metrics, specially recall and F1 scores are not promising. Subsequently, the RoBERTa sentence transformer model fine-tuned on CosineSimilarityLoss in conjunction with XGB classifier achieves the highest accuracy of 88.4%. It is also observed that finetuning the RoBERTa sentence transformer model on CosineSimilarityLoss provides a lift of 4.2 percentage points in terms of accuracy, a lift of $> 9$ percentage points in terms of recall and a lift of $> 5$ percentage points in terms of F1 score for the minority classes (positive and neutral sentiment classes) when compared to the performance of the corresponding baseline sentence transformer model. This demonstrates the contribution of the fine-tuning process for sentence transformer models in enhancing the models' classification performance, particularly for the minority classes. Subsequently, the best-performing fine-tuned model from the Twitter US Airline Sentiment dataset was evaluated on the entire Twitter Apple Sentiment dataset, achieving an impressive accuracy of 88.5% in a tri-class sentiment classification scenario.

For the IMDB dataset, finetuning the RoBERTa sentence transformer model on CosineSimilarityLoss provides a lift of 5.4 percentage points in terms of accuracy over the performance of baseline sentence transformer model. Subsequently, the fine-tuned model from the IMDB dataset was evaluated on the Yelp test dataset, achieving an impressive accuracy of 94.8% in a binary sentiment classification scenario.

So, in both the above cases, we observe the fine-tuned models were able to retain their generalization capabilities with decent accuracies without the need of any additional supervised fine-tuning when evaluated on different datasets across different domains.

Figure 4 displays the normalized confusion matrix for the finetuned model with the highest accuracy on Twitter US Airline Sentiment, thus indicating the recall rates diagonally in blue boxes for each of the classes.

Table 4 presents a comparative analysis of our methodology and Meta-Llama-3-8B (AI@Meta, 2024) in terms of performance, training time, and inference time for the Twitter US Airline Sentiment and IMDB datasets under both zeroshot and fine-tuned settings. We have fine-tuned the pre-trained version of Meta-Llama-3-8B on the respective training datasets for performance comparison with our fine-tuned models.

Table 4: Our Methodology vs Meta-Llama-3-8B

| Dataset | Model | Accuracy | Training time | Inference time |
|---------|-------|----------|---------------|----------------|
| TAS | Meta-Llama-3-8B - zeroshot | 48.0% | N/A | 4m |
| TAS | Meta-Llama-3-8B - fine-tuned | 85.9% | 2 hr 47m[5] | 4m |
| TAS | Our model [6] | 88.4% | 15m | 13m |
| IMDB | Meta-Llama-3-8B - zeroshot | 50.2% | N/A | 2hr 48m |
| IMDB | Meta-Llama-3-8B - fine-tuned | 97.1% | 7hr 11m[7] | 2hr 51m |
| IMDB | Our model [8] | 95.9% | 4hr 23m | 11m |

From Table 4, it is evident that the fine-tuned Meta-Llama-3-8B model achieves an accuracy of 85.9% on the Twitter US Airline Sentiment dataset, which is lower than the accuracy achieved by our fine-tuned sentence transformer model by 2.5 percentage points. Conversely, the Meta-Llama-3-8B model attains an accuracy of 97.1% on the IMDB dataset, surpassing the performance of our fine-tuned sentence transformer model by 1.2 percentage points. However, for both datasets, the total training time under supervised fine-tuning is significantly lower for the sentence transformer models compared to the Llama models. Specifically, for

---

[5]No. of epochs = 3, train:val:test = 60:20:20, fine-tuning method - Quantized Low-Rank Adaptation (QLoRA)(Mamba, 2024)

[6]RoBERTa-Large ST + CosineSimilarityLoss + XGB

[7]No. of epochs = 1, train samples:20,000, val samples:5000, test samples:25,000, fine-tuning method - QLoRA

[8]RoBERTa-Large ST + CosineSimilarityLoss + SVM

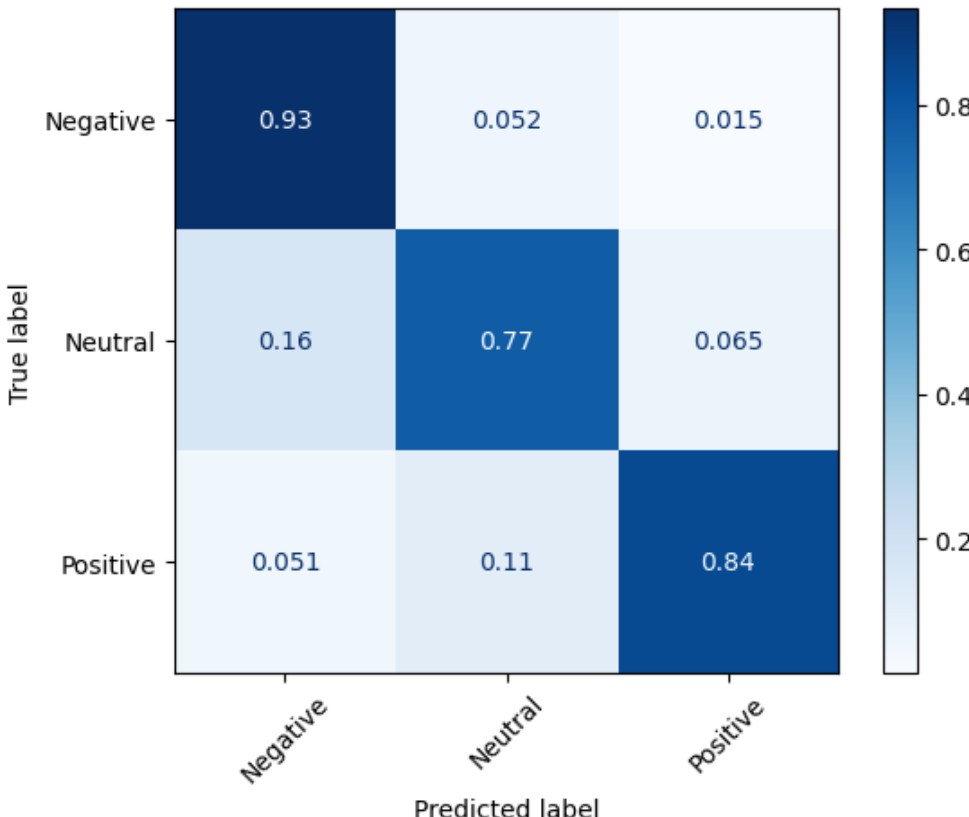

Figure 4: Normalized confusion matrix for Twitter US Airline Sentiment Data

the Twitter US Airline Sentiment dataset, using the L4 GPU with 24 GB of memory, the Meta-Llama-3-8B model's training time is 11 times longer than that of the sentence transformer model. For the IMDB dataset, using the A100 GPU with 40 GB of memory, the Meta-Llama-3-8B model's training and inference times are 1.6 times and 15 times longer, respectively, compared to the sentence transformer model.

## 5  Broader Impact Statement

Sentiment analysis is a powerful technique in artificial intelligence that has important business applications. For example, organizations can use sentiment analysis to gauge public opinion about their products and services. E-commerce companies can monitor the overall sentiment of reviews of their different products to identify any suspicious vendors that use false advertising or scamming tactics. This research can help businesses develop inexpensive, easy to implement models for sentiment analysis with state-of-the-art performance. Below are some perceived advantages of our approach over existing techniques for sentiment classification.

**Advantages over existing techniques**

- **No text cleaning**

  One of the key challenges faced by existing methods using the conventional text vectorization methods like Bag-of-Words and TF-IDF is the need for extensive text preprocessing (such as removing stopwords, special characters, and emojis). This not only makes the entire process computationally more expensive but also leads to loss of valuable information. On the other hand, sentence transformers are designed to handle raw text effectively. They learn contextual embeddings directly

from the input text. Unlike the traditional text vectorization methods, sentence transformers capture semantic meaning, making them robust to minor noise (e.g., typos, punctuation, and special characters).

- **Lesser computational cost**

  Techniques leveraging transformer architectures like BERT or GPT-based large language models may provide comparable or marginally better performances for sentiment classification but they usually come with a high computational price tag. Sentence transformers directly produce embeddings for entire sentences, enabling them to handle variable-length input (sentences) without increasing computational complexity which is why their processing is more efficient than transformer models' token-by-token processing. As a result, sentence transformers are generally less computationally expensive under both pre-trained and supervised fine-tuned settings.

- **Preservation of generalizability**

  While fine-tuning can narrow a model's focus and impair its ability to generalize across different tasks or domains, fine-tuned sentence transformer models maintain their generalizability. This is due to the robust foundation of the underlying transformer models and the use of appropriate loss functions that minimize losses. As a result, these models are well-suited for classification tasks across various domains, even when encountering data that deviates from their fine-tuned training set, without significant loss in generalizing ability.

  **Limitations**

  Like any fine-tuning strategy, a notable limitation of our methodology is fine-tuning can introduce biases present in the training data into the model. This inherited bias, when replicated by machine learning models, can result in unfair decisions, particularly in tasks involving identity dimensions such as race and gender. In such cases, metrics calculated on the entire test set may obscure these biases. A potential mitigation strategy is to evaluate subsets (e.g., minority groups) separately.

The future plan is to extend this study further to incorporate text summarization and topic modeling for the review texts whose sentiments will be predicted by the model so as to identify the major keywords responsible for such predicted sentiments, especially for the negatively predicted sentiments. This will enable organizations to recognize their own shortcomings on a real time basis just by ingesting the customer reviews.

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
