# OpenReview forum: "Sentiment Classification using Sentence Embeddings: Exploiting Sentence Transformer Loss Functions"
_TMLR — Rejected by TMLR_

### Review · Reviewer_oAbp · 2024-09-30

**Summary Of Contributions:**

The paper proposes a sentiment classification method leveraging sentence embeddings from a pre-trained transformer (mpnet-base) fine-tuned using CosineSimilarity loss. These embeddings are then combined with machine learning models like LightGBM and XGBoost for sentiment classification on the “Twitter US Airline Sentiment” dataset. The method reports an accuracy of 86.5% with SOTA results.


The contributions are:

1. The paper introduces a sentiment classification method using sentence embeddings generated by a pre-trained transformer model (mpnet-base) fine-tuned with CosineSimilarity loss.


2. The proposed model demonstrates improved recall for minority sentiment classes (74.4% for neutral and 82.9% for positive) without requiring data augmentation, addressing the issue of class imbalance in the dataset.

3. The paper provides a detailed exploration of various loss functions (e.g., CosineSimilarityLoss, CoSENT, and TripletLoss) and their impact on fine-tuning the sentence transformer model for sentiment classification.

**Audience:**

Yes

**Broader Impact Concerns:**

The broader impact statement lacks depth and could discuss more significant concerns such as model bias and fairness.

**Claims And Evidence:**

Yes

**Requested Changes:**

1. Further Exploration of Baselines: The model’s performance should be compared against newer sentiment classification techniques, particularly modern transformer models like RoBERTa or GPT-based approaches. These comparisons would provide a more grounded understanding of whether the proposed method truly advances the state of the art​.

2. Broader Dataset Evaluation: The authors should test their model on more diverse and up-to-date datasets to prove its generalizability.

**Strengths And Weaknesses:**

**Strengths**

1. Fine-Tuning Strategy: The paper explores various loss functions (e.g., CosineSimilarityLoss, TripletLoss, CoSENT) for fine-tuning, which provides an interesting contribution to the understanding of loss function impacts on sentiment classification.


2. Minority Class Performance: The method shows improvements in recall for minority classes (74.4% for neutral, 82.9% for positive), which is notable given the imbalanced nature of the dataset​

**Weaknesses**

1. Unfair Comparison Due to Pre-Trained Model: The paper compares the performance of the proposed fine-tuned model against models like Random Forest and other simpler architectures without acknowledging the unfair advantage gained by using a pre-trained transformer (mpnet-base). For example, in Table 1, the paper claims a higher accuracy (86.5%) than most baselines, but this is primarily because other models, like Random Forest, do not benefit from any form of pre-training. Despite claiming superiority, the proposed method does not outperform all baseline methods. For example, a previous study using Word2Vec with RNN achieves 90% accuracy in binary classification.

2. Generalization of datasets: The authors only evaluate their model on the outdated “Twitter US Airline Sentiment” dataset, collected in 2015. This dataset may not represent current sentiment trends or language usage.

---

> ### Author Response · Authors · 2024-09-30
> **Response to reviewer**
>
> We appreciate the time and effort that the reviewer has dedicated to providing valuable feedback on our manuscript. We are grateful to the reviewer for the insightful comments on our paper. Here is a point-by-point response to the reviewer's comments and concerns.
>
> Weakness 1 - Unfair Comparison Due to Pre-Trained Model: The paper compares the performance of the proposed fine-tuned model against models like Random Forest and other simpler architectures without acknowledging the unfair advantage gained by using a pre-trained transformer (mpnet-base). For example, in Table 1, the paper claims a higher accuracy (86.5%) than most baselines, but this is primarily because other models, like Random Forest, do not benefit from any form of pre-training. Despite claiming superiority, the proposed method does not outperform all baseline methods. For example, a previous study using Word2Vec with RNN achieves 90% accuracy in binary classification.
>
> Response – Agreed that the paper compares performance of our model with those models as well which do not rely on the advanced transformer-based architectures or pre-trained models. However, the intention was purely to show how the sentiment classification SotA techniques have evolved over time.
>
> Authors in Dang et al. 2020, who had demonstrated 90% accuracy, had removed all samples labeled “neutral”, leaving only two classes for the experiment, i.e. positive and negative. You would appreciate the fact that it is always more challenging to accurately classify three classes (positive, neutral, and negative) than just two (positive and negative). We are not sure how their model would have responded to tri-class sentiment scenario, what would have been the accuracy based on three sentiment classes and more importantly, what would have been the recall rate for the neutral class since it is always trickier to correctly classify the neutral class sentiments.
>
> Weakness 2 - Generalization of datasets: The authors only evaluate their model on the outdated “Twitter US Airline Sentiment” dataset, collected in 2015. This dataset may not represent current sentiment trends or language usage.
>
> Response – Agreed. We are willing to conduct experiments on more recent dataset and will revert with the results as early as possible.
>
> Requested Changes 1 - Further Exploration of Baselines: The model’s performance should be compared against newer sentiment classification techniques, particularly modern transformer models like RoBERTa or GPT-based approaches. These comparisons would provide a more grounded understanding of whether the proposed method truly advances the state of the art.
>
> Response – We have outlined one reference in table 1 (Talaat, 2023) where the author has used a hybrid architecture of RoBERTa with BiGRU layers which yielded the highest accuracy of 86% for the US airlines dataset. Nevertheless, we will try to come up with more references where newer sentiment classification techniques have been used for this dataset.
>
> Requested Changes 2 - Broader Dataset Evaluation: The authors should test their model on more diverse and up-to-date datasets to prove its generalizability.
>
> Response – Agreed. We are willing to conduct experiments on more recent and diverse dataset and will revert with the results as early as possible.
>
> Broader Impact Concerns - The broader impact statement lacks depth and could discuss more significant concerns such as model bias and fairness
>
> Response – Agreed. We plan to incorporate necessary modifications to address this issue and will revert as soon as possible.

---

### Review · Reviewer_hjFX · 2024-10-07

**Summary Of Contributions:**

This paper tackles the sentiment classification problem using sentence embeddings learned by neural networks. The main experiments are conducted on the "Twitter US Airline Sentiment" dataset to leverage pre-trained transformer architectures, loss functions for fine-tuning and further combined with classical machine learning models for inference. The experimental results suggest that the cosine similarity loss function with LightGBM as a classification model yields the best performance.

**Audience:**

No

**Claims And Evidence:**

Yes

**Requested Changes:**

Combining the reasons I listed in the weaknesses, I think this paper is below the bar of TMLR and may require even more changes than a major revision.

**Strengths And Weaknesses:**

Strengths:
* The paper is well-written, and the main idea is very easy to follow.
* Studying how different architecture combinations and loss functions over the entire machine learning pipeline is important and could lead to new insights and methodologies.

Weaknesses:
* This paper mainly does a comparative study of different components in sentiment analysis, the discussions on experimental results are short and shallow.
* Question: do you know if the tasks have been included in the pre-training dataset of MPNet-base?
* Despite the literature on sentiment analysis seems to be less studied through the years (Table 1), the improvement of the proposed method is very limited.
* I am uncertain if the paper would be of interest to TMLR or the general machine learning community because most of the references regarding this problem (See in Sec 2) are either from several years ago (and/or only on arXiv), or published in journals outside of ML. Even though I think this paper provides a comparative study of machine learning methods on sentiment analysis, my personal feeling is that it is better suited to some journals where people need tools for it.

---

> ### Author Response · Authors · 2024-10-12
> **Response to Reviewer hjFX's comments**
>
> We appreciate the time and effort that the reviewer has dedicated to providing valuable feedback on our manuscript. We are grateful to the reviewer for the insightful comments on our paper. Here is a point-by-point response to the reviewer's comments and concerns.
>
> **Weakness 1:**
>
> This paper mainly does a comparative study of different components in sentiment analysis, the discussions on experimental results are short and shallow.
>
> **Response:**
>
> Since this is a classification problem, we have focused on the standard metrics like accuracy, precision, recall and F1 score for the different models in the results section. We have also highlighted how the recall rates of the minority classes improved by using the fine-tuned transformer models without any data augmentation.  We have included the normalized confusion matrix in the results section as a ready reference of the performance metrics achieved by our best model. We also highlighted the lift we are getting in terms of accuracy, recall rates and F1 scores when using the fine -tuned models in comparison to the baseline models.
>
> **Weakness 2:**
>
> Question: do you know if the tasks have been included in the pre-training dataset of MPNet-base?
>
> **Response:**
>
> Authors introducing Mpnet in their paper “Mpnet: Masked and permuted pre-training” has stated that the model was pre-trained on a large text corpus following the lines of XLNet and RoBERTa like BOOKCORPUS plus English WIKIPEDIA, CC-NEWS, OPENWEBTEXT, STORIES, etc. Till now, we have not found any reference indicating MPNet was pre-trained/fine-tuned on the Twitter Airline Sentiment dataset. This is to the best of our knowledge.
>
> **Weakness 3:**
>
> Despite the literature on sentiment analysis seems to be less studied through the years (Table 1), the improvement of the proposed method is very limited.
>
> **Response:**
>
> Through our methodology, we have demonstrated that sentence transformer embeddings can be semantically rich and can deliver excellent performances for sentiment classification, even for minority classes in an imbalanced dataset, without requiring data augmentation. Additionally, we wanted to highlight through our paper that, although baseline sentence transformers provide less granular embeddings compared to transformer models like BERT or RoBERTa but fine-tuned sentence transformer models combined with robust machine learning algorithms can match or sometimes even surpass the performance of transformer models, albeit with reduced computational load. So, we want to emphasize that while the improvements from the proposed method may not always seem groundbreaking in terms of performance metrics, they offer significant advantages in efficiency, ease of use, and reduced computational power. These benefits make the method particularly suitable for deployment in resource-constrained environments.
>
> **Weakness 4:**
>
> I am uncertain if the paper would be of interest to TMLR or the general machine learning community because most of the references regarding this problem (See in Sec 2) are either from several years ago (and/or only on arXiv), or published in journals outside of ML. Even though I think this paper provides a comparative study of machine learning methods on sentiment analysis, my personal feeling is that it is better suited to some journals where people need tools for it.
>
> **Response:**
>
> We will provide more references which have been recently published and /or published in renowned journals/conference proceedings in our next revision under table 1 of section 2 of our manuscript.

---

### Review · Reviewer_y86U · 2024-10-14

**Summary Of Contributions:**

The paper presents a new approach for performing sentiment classification. The proposed approach uses a Transformer model (mp-net) and is trained using cosine similarity loss and gradient boosting. The proposed approach outperforms the mentioned baselines to achieve the best performance on Twitter Airlines dataset.

**Audience:**

Yes

**Claims And Evidence:**

Yes

**Requested Changes:**

1. Comparison with state-of-the-art open-sourced and closed-sourced models (like GPT-4 APIs, Llama 3 70B, Llama 3 8B, etc.) in both zero-shot and fine-tuned settings.
2. Experiments on more sentiment classification datasets like SST, Amazon Reviews, Yelp Reviews, etc. The trained models should also be evaluated on the test set of other datasets to observe the generalization capabilities.
3. Comparison of the time complexity (both training and inference) of the proposed approach compared with baseline approaches present in the paper and suggested baselines.

**Strengths And Weaknesses:**

**Strengths**:

1. The paper is fairly easy to understand.


**Weaknesses**:

1. The novelty of the proposed approach is unclear. The cosine similarity loss and GBM are studied in the literature. The current work just reuses these techniques for the task of sentence classification.
2. The paper ignores the latest baselines using stronger pre-trained or instruction-tuned models like Llama 3 and GPT-4 in both zero-shot and fine-tuned settings.
3. After comparison with the above baselines, this work should compare the advantages of the proposed approach in terms of execution or training time.
4. The paper should also include experiments on other datasets like SST, Amazon Reviews, Yelp Reviews, etc. The paper should also test the generalization ability of the trained model (e.g., train a model on SST and test on Amazon, etc.).

**Minor comments**:

unspecified citation: Tusar & Islam (2021)

Table 1: the paper should report the results of all methods using the same precision (e.g., correct to 2 decimal places

“sentence embeddings which are nothing but compressed information along a sequence of token embeddings with lower level of granularity.” — unclear what this means.

“ finding proximity among  ****those embeddings is now computationally much less intensive and achieves similar results when compared to transformer models like BERT or RoBERTa.”

---

> ### Author Response · Authors · 2024-10-26
> **Response to comments by Reviewer y86U**
>
> **Weakness 1:**
>
> The novelty of the proposed approach is unclear. The cosine similarity loss and GBM are studied in the literature. The current work just reuses these techniques for the task of sentence classification.
>
> **Response**
>
> It is acknowledged that cosine similarity loss and GBM are well-explored techniques. The novelty of our approach lies in the integration of these methods within the context of sentiment classification. While these techniques have been individually studied, our paper explores their combined efficacy that previous works have not fully explored. In particular, we demonstrate that fine-tuned sentence transformers can deliver excellent performance for sentiment classification, even for minority classes in imbalanced datasets, without requiring data augmentation. Additionally, our paper highlights that fine-tuned sentence transformer models combined with robust machine learning algorithms can match or surpass the performance of transformer models, with reduced computational load.
> We emphasize that while the improvements from our proposed method may not always seem groundbreaking in terms of performance metrics, they offer significant advantages in ease of use and reduced computational power. These benefits make the method particularly suitable for deployment in resource-constrained environments.
>
> **Weakness 2:**
>
> The paper ignores the latest baselines using stronger pre-trained or instruction-tuned models like Llama 3 and GPT-4 in both zero-shot and fine-tuned settings.
>
> **Response**
>
> We have incorporated performances of pre-trained advanced models like Meta-Llama-3-8B (refer table 4) and Flan 137B (refer table 1) in our paper
>
> **Weakness 3:**
>
> After comparison with the above baselines, this work should compare the advantages of the proposed approach in terms of execution or training time.
>
> **Response**
>
> Please refer table 4 of our paper where we have provided a comparative study between our proposed approach and Meta-Llama-3-8B in terms of performance, training, and inference time under both zero-shot and finetuned settings.
>
> **Weakness 4:**
>
> The paper should also include experiments on other datasets like SST, Amazon Reviews, Yelp Reviews, etc. The paper should also test the generalization ability of the trained model (e.g., train a model on SST and test on Amazon, etc.).
>
> **Response**
>
> To evaluate the robustness of our methodology, we have done the following:
>
> 1.	Used another dataset “IMDB” and adopted similar approach to develop a fine-tuned sentence transformer model based on Roberta-Large as the underlying transformer model. The results are tabulated in table 3 of the paper.
>
> 2.	The above fine-tuned and trained model from IMDB training dataset was further evaluated on Yelp-2 test dataset to demonstrate our model’s generalizability under a binary sentiment classification scenario and the results are tabulated in table 3 of the paper.
>
> 3.	In addition to the MPNet transformer model, we have also developed a separate sentence transformer model based on Roberta-Large transformer model for our initial dataset “Twitter US Airline Sentiment” and the results are tabulated in table 3 of the paper.
>
> 4.	We used another dataset “Twitter Apple Sentiment” on which we evaluated our model fine-tuned and trained from our initial dataset “Twitter US Airline Sentiment” to demonstrate the model’s generalizability in a tri-class sentiment classification scenario and the results are tabulated in table 3 of the paper.
>
> **Minor comments 1**
>
> unspecified citation: Tusar & Islam (2021)
>
> **Response**
>
> We have removed this citation and relevant text from our paper to focus only on the studies employing more advanced techniques.
>
> **Minor comments 2**
>
> Table 1: the paper should report the results of all methods using the same precision (e.g., correct to 2 decimal places)
>
> **Response**
>
> Corrected to 1 decimal places.
>
> **Minor comments 3**
>
> “sentence embeddings which are nothing but compressed information along a sequence of token embeddings with lower level of granularity.” — unclear what this means.
>
> **Response**
>
> We have rephrased the above-mentioned text under 3.3.1 of our paper for better clarity.
>
> **Requested Changes 1:**
>
> Comparison with state-of-the-art open-sourced and closed-sourced models (like GPT-4 APIs, Llama 3 70B, Llama 3 8B, etc.) in both zero-shot and fine-tuned settings.
>
> **Response**
>
> Please refer to response under weakness 2
>
> **Requested Changes 2:**
>
> Experiments on more sentiment classification datasets like SST, Amazon Reviews, Yelp Reviews, etc. The trained models should also be evaluated on the test set of other datasets to observe the generalization capabilities.
>
> **Response**
>
> Please refer to response under weakness 4
>
> **Requested Changes 3:**
>
> Comparison of the time complexity (both training and inference) of the proposed approach compared with baseline approaches present in the paper and suggested baselines.
>
> **Response**
>
> Please refer to response under weakness 3

---

### Decision · Action_Editor_BXqb · 2024-12-05

**Recommendation:** Reject

**Comment:**

The paper investigates a method for sentiment classification that leverages sentence embeddings that are produced from fine tuning a pre-trained transformer model.

The reviewers note that this paper does not introduce genuinely novel ideas, does not report results that significantly outperform the state of the art, does not investigate the results in sufficient detail, and does not compare to an adequate selection of baselines and datasets. As a result, the claims and evidence do not seem to provide adequate support for the conclusions, and since the topic and approach are significantly out of date, it is unlikely that the current version of the paper will be of interest to members of the TMLR community.

**Audience:**

It is unlikely that more than a select few individuals would be interested in the findings of this paper.

**Claims And Evidence:**

No, the evidence is not sufficiently convincing.